# Central Roles of STAT3-Mediated Signals in Onset and Development of Cancers: Tumorigenesis and Immunosurveillance

**DOI:** 10.3390/cells11162618

**Published:** 2022-08-22

**Authors:** Shigeru Hashimoto, Ari Hashimoto, Ryuta Muromoto, Yuichi Kitai, Kenji Oritani, Tadashi Matsuda

**Affiliations:** 1Division of Molecular Psychoimmunology, Institute for Genetic Medicine, Hokkaido University, Sapporo 060-0815, Japan; 2Department of Molecular Biology, Graduate School of Medicine, Hokkaido University, Sapporo 060-8638, Japan; 3Department of Immunology, Graduate School of Pharmaceutical Sciences, Hokkaido University, Sapporo 060-0812, Japan; 4Department of Hematology, International University of Health and Welfare, Narita 286-8686, Japan

**Keywords:** STAT3, tumorigenesis, immune evasion, STAP-2, ARID5A

## Abstract

Since the time of Rudolf Virchow in the 19th century, it has been well-known that cancer-associated inflammation contributes to tumor initiation and progression. However, it remains unclear whether a collapse of the balance between the antitumor immune response via the immunological surveillance system and protumor immunity due to cancer-related inflammation is responsible for cancer malignancy. The majority of inflammatory signals affect tumorigenesis by activating signal transducer and activation of transcription 3 (STAT3) and nuclear factor-κB. Persistent STAT3 activation in malignant cancer cells mediates extremely widespread functions, including cell growth, survival, angiogenesis, and invasion and contributes to an increase in inflammation-associated tumorigenesis. In addition, intracellular STAT3 activation in immune cells causes suppressive effects on antitumor immunity and leads to the differentiation and mobilization of immature myeloid-derived cells and tumor-associated macrophages. In many cancer types, STAT3 does not directly rely on its activation by oncogenic mutations but has important oncogenic and malignant transformation-associated functions in both cancer and stromal cells in the tumor microenvironment (TME). We have reported a series of studies aiming towards understanding the molecular mechanisms underlying the proliferation of various types of tumors involving signal-transducing adaptor protein-2 as an adaptor molecule that modulates STAT3 activity, and we recently found that AT-rich interactive domain-containing protein 5a functions as an mRNA stabilizer that orchestrates an immunosuppressive TME in malignant mesenchymal tumors. In this review, we summarize recent advances in our understanding of the functional role of STAT3 in tumor progression and introduce novel molecular mechanisms of cancer development and malignant transformation involving STAT3 activation that we have identified to date. Finally, we discuss potential therapeutic strategies for cancer that target the signaling pathway to augment STAT3 activity.

## 1. Introduction

The importance of chronic inflammation in the mechanism of cancer has been well-established [1]. Inflammation plays a crucial role in almost all aspects of the tumorigenic process [2]. The role of inflammation in tumorigenesis has been extensively investigated, and recent lines of evidence provide a possible links between inflammation and tumor recurrence and metastasis [2,3]. However, the effects of inflammatory signaling on tumorigenesis remain elusive.

In 1994, signal transducer and activation of transcription 3 (STAT3) was discovered as a transcription factor involved in interleukin-6 (IL-6)-induced hepatic acute phase responses by Kishimoto and Akira’s group and Darnell’s group [4,5]. IL-6, the best-known protumor cytokine, and its family of cytokines, including IL-11, IL-27, IL-31, cardiotrophin-1, ciliary neurotrophic factor, leukemia inhibitory factor (LIF), and oncostatin M (OSM), are involved in crucial physiological and/or pathological processes, such as cell growth, survival, differentiation, energy metabolism, angiogenesis, migration, invasion, metastasis, inflammation, and autoimmune diseases [6,7,8,9,10]. The IL-6 family cytokines, excluding IL-31, can transduce intracellular signals linking with the Janus kinase (JAK)-STAT3 pathway, the Src homology 2 (SH2)-containing protein tyrosine phosphatase-2 (SHIP2)-Ras-Raf-MEK-extracellular signal-regulated kinase (ERK) pathway, and the phosphoinositide 3-kinase (PI3K)-Akt pathway mediated by the activation of shared signal-transducing receptor component glycoprotein 130 (gp130, IL6ST) [6,7,8,9,10]. In these signaling pathways, STAT3 is considered to be a key signaling molecule of the IL-6-gp130 pathway because it acts as an oncogenic driver and plays an important role in mediating tumor-promoting inflammation [6,7,9,11,12]. Importantly, suppressor of cytokine signaling 3 (SOCS3) is induced by STAT3 and is postulated to modulate the primary negative regulation of the gp130-mediated signaling pathway [13,14].

Besides the IL-6 family, activation of cellular STAT3 is also triggered by hepatocyte growth factor receptor, c-MET, epidermal growth factor (EGF) receptor (EGFR) [15,16], and Src family kinases [11,12]. Furthermore, it has been reported that G-protein-coupled receptors, such as sphingosine-1-phosphate receptor 1, stimulate STAT3 via JAK and Src family kinases [17,18] and that Toll-like receptors (TLRs), such as TLR9 and TLR4, are considered to play crucial roles in inflammation via the activation of the JAK-STAT3 pathway [19,20,21]. MicroRNAs (miRNAs) such as miR-17-5p, miR-20a, miR-124, and miR-551b-3p have emerged as key modulators of cancer biology, and some of these miRNAs have been shown to be pivotal in the regulation of the JAK-STAT3 pathway [22,23,24,25,26] (Figure 1).

Moreover, STAT3 not only contributes functionally to promoting tumor cell proliferation, survival, invasion, angiogenesis, and immune evasion, but it has been recently indicated to play crucial roles in the inflammation associated with tumorigenesis, obesity and metabolic syndrome, the cancer stemness pathway, and premetastatic niche formation [11,12,27,28,29,30,31,32]. Furthermore, STAT3 has been shown to be involved in the formation of immunosuppressive tumor microenvironments (TMEs) via regulating not only immune cells but also cancer-associated fibroblasts (CAFs) and endothelial cells.

We previously performed a series of studies analyzing the roles of signal-transducing adaptor protein-2 (STAP-2) in the proliferation of several types of cancer cells via acting as an adaptor molecule that modulates STAT3 activity [33] and recently found that AT-rich interactive domain-containing protein 5A (ARID5A) functions as an RNA-binding molecule that stabilizes mRNAs such as those of indoleamine 2,3-dioxygenase 1 (IDO1), C-C motif chemokine ligand 2 (CCL2), and STAT3, resulting in the induction of an immunosuppressive TME in malignant tumors [34]. In this review, we provide an overview of the recent findings regarding the intrinsic and extrinsic roles of STAT3 during tumor progression. We further introduce the novel molecular mechanisms that we have identified to date involving STAT3 activation, cancer development, and malignant transformation. Finally, we address the potential therapeutic strategies against malignant tumors by targeting the signaling pathway to augment STAT3 activity.

## 2. STAT3 Signal in Cancer Cells

IL-6 is a well-established tumor-promoting cytokine among the IL-6 family of cytokines, which activates multiple STAT3-mediated tumor initiation and progression pathways [11,12,35,36]. For example, the IL-6/STAT3 axis enhances the transcriptional activation of various molecular targets that are crucial for cell cycle progression and survival (e.g., cyclin D1, myc, Bcl2-like 1, survivin, and miR-21) and angiogenesis (e.g., hypoxia-inducible factor 1α, vascular endothelial growth factor (VEGF), and matrix metalloproteinases (MMPs, e.g., MMP2, MMP7, and MMP9)) [37,38,39]. In the late stages of cancer, IL-6/STAT3 may promote the gain of invasive activity and the metastatic dissemination of cancer cells by inducing epithelial–mesenchymal transition (EMT) transcription factors (EMT-TFs), such as SNAI1 and TWIST [40]. Notably, the EMT program in cancer biology has been implicated to facilitate not only cell motility and invasiveness but to also possibly be involved in cancer stem cell (CSC) status and the resistance to anticancer drugs via epithelial–mesenchymal plasticity [41,42,43]. Consistently, IL-6/STAT3 signaling in cancerous EMT also results in the acquisition of cancer stemness in cancer cells; the self-renewal and population expansion of CSCs requires STAT3 in cooperation with stem-cell-associated transcription factors, such as NANOG [44]. In addition, IL-6/STAT3 signaling is crucial for the shift from non-CSCs to CSCs by upregulating the expression of Oct4 [45]. These functions of IL-6/STAT3 signaling ultimately lead to the development of several multidrug-resistant and malignant phenotypes [44,45] (Figure 1).

### 2.1. Pancreatic Cancer

High serum levels of IL-6 have been associated with poor overall survival prognosis in patients with highly malignant pancreatic cancer [46], and increased activity of IL-6/STAT3-mediated signaling has been reported to be associated with poor prognosis in patients with pancreatic ductal adenocarcinoma (PDAC) after resection [47]. The activation of STAT3 in PDAC has been reported in patient-derived clinical specimens and pancreatic cancer cells [48] and is a prognostic risk factor [49]. IL-6 also induces a mesenchymal phenotype in human pancreatic cancer cells via STAT3 activation and SNAI1 induction [50]. Interestingly, in a mouse model, STAT3 is involved in the reprogramming of acinar-to-ductal metaplasia (ADM), which is triggered by the sustained exocrine-tissue-specific expression of pancreatic and duodenal homeobox 1 (Pdx1), which is a pancreatic-progenitor-specific transcription factor [51]. ADM transdifferentiation occurs in chronic pancreatitis via STAT3 and is associated with pancreatic intraepithelial neoplasia (PanIN), which is a necessary step for the generation of neoplastic precursor lesions [52,53]. In a PDAC mouse model driven by KRAS [52,53], for instance, it has been reported that pancreatic epithelial cells bearing the constitutively active *KRAS* mutation KRAS^G12D^ trigger inflammation activation by recruiting immune cells, particularly myeloid cells, that facilitate the production of IL-6 and soluble IL6R (sIL6R) and, in turn, activate STAT3 via IL-6 trans-signaling through the binding of IL-6 to the soluble form of IL6R and the subsequent binding of IL-6 and sIL6R complexes to gp130-expressing cells [52]. Dysregulated STAT3 activation due to the homozygous loss of *SOCS3* in the pancreas leads to the accelerated progression of PanIN and the onset of PDAC [52]. It has also been shown that the activation of KRAS increases cytokine levels, such as IL-6 and IL-11 in epithelial cells, which subsequently drives STAT3 activation in an autocrine manner, and that STAT3-triggerd MMP7 is necessary for tumor progression but not for tumor onset, which might be regulated by other STAT3 targets [53].

Because of the TME of PDAC, in which the low vascular density results in severe hypoxia and limited nutrient utilization, PDAC cells are known to have increased autophagy to rewire their metabolism to survive and maintain metabolic homeostasis in harsh environments [54,55]. In the mouse model of PDAC caused by *KRAS* mutations, increased levels of autophagy are required for IL-6-induced STAT3 activation. Mechanistically, the receptor for advanced glycation products promotes the IL-6-driven activation of STAT3 signaling in mitochondria, providing a bridge between autophagy and the IL-6-STAT3 signaling pathway [56].

### 2.2. Colorectal Cancer

Increased levels of IL-6 and sIL6R in the circulating blood and intestine in patients with inflammatory bowel disease are primary risk factors for colitis-associated cancer (CAC) [57]. The serum levels of IL-6 in patients with colorectal cancer (CRC) are correlated with the malignant tumor grade, and high IL-6 levels (≥10 pg/mL) are an independent indicator of poor prognosis [58]. Controversially, although it has also been shown that the IL-6 levels in the serum correlate with disease progression in CRC patients, the IL-6 level is not an independent prognostic marker [59]. On the other hand, the activation of STAT3 has been reported to associate with poor outcomes in CRC patients [60,61,62]. The expression of both IL6R and gp130 have been observed in epithelial cells of the intestine and in immune cells, and the release of membrane-bound IL6R has been detected in the serum with the progression of CAC [63]. sIL6R released within the TME can induce STAT3 activation in gp130-expressing cells by trans-signaling [64].

Compositional changes in the microbiota are associated with a predisposition to the development of colorectal tumors. It has been demonstrated that a high intake of dietary fat and meat is associated with a high risk of colorectal cancer, which may result from diet-induced differences in the microbiota composition and metabolic activities [65]. It has also been shown, using a CAC mouse model created by the injection of azoxymethane (AOM) followed by treatment with dextran sulfate sodium salt (DSS), that apoptosis-associated speck-like protein containing a caspase recruitment domain or NOD-like receptor family pyrin domain containing 6 performs important functions in CAC progression. Furthermore, an interesting finding has been observed in that wild-type mice cohabitating with mice lacking these inflammasome genes are more vulnerable to the initiation of CAC [66]. Mechanistically, IL-18-induced changes in the microbiota induce CC-chemokine ligand 5-driven inflammation, which accelerates epithelial cell proliferation through the regional activation of the IL-6/STAT3 pathway, eventually resulting in cancer formation [66].

Although the signal transduction of IL-6 is crucial for STAT3 activation in CRC initiation and development [63,67,68,69], the ablation of STAT3 in intestinal enterocytes has more significant effects on mucosal damage and regeneration, tumor growth, and proliferation than the lack of IL-6 in the CAC model induced by AOM and DSS [68], indicating that other cytokines involved in STAT3 activation, such as EGF family growth factors, IL-11, and IL-22, as well as hormones, such as leptin, may drive the activation of STAT3 in inflammation-induced CRC cells.

Sporadic CRC in colorectal adenomatosis Apc (Min/+) mice, a commonly used animal model bearing numerous adenomatous polyps reflecting familial adenomatosis of the colon based on heterozygosity for *Apc* truncation mutations, is also essential for IL-6 signaling [70,71]. Apc (Min/+) mice lacking *STAT3* had a reduced occurrence of and suppressed the growth of early adenomas [72]. However, STAT3 deficiency promoted late tumor progression and led to the formation of invasive and metastatic carcinomas via the enhancement of carcinoembryonic antigen-related cellular adhesion molecule 1, which is involved in intercellular adhesion [72]. Conversely, it has also been suggested that STAT3 does not affect tumorigenesis, but the downregulation of Snail1 inhibits the transition from adenoma to cancer in Apc (Min/+) mice [73]. Additionally, IL-11 has been shown to correlate more strongly than IL-6 with increased STAT3 activation in human CRC specimens [70]. Subsequently, it has been demonstrated that IL-11/STAT3-mediated signaling functions as a stronger promoter of the progression of sporadic and inflammation-associated CRC than IL-6/STAT3 signaling in the progression of sporadic and inflammation-associated CRC progression, suggesting that IL-11/STAT3 signaling is a promising therapeutic target for the cure of CRCs [70,74]. Notably, the results of the possible tumor-suppressive roles of STAT3 in a CRC mouse model require further investigation regarding the underlying molecular mechanisms and consistency with clinical observations.

IL-6, together with transforming growth factor beta (TGF-β), induces the generation of Th17 cells, and Th17 cells and other cells producing IL-17A trigger sporadic CRC in mice and humans [75,76,77]. The “Th17 gene expression profiling” in stage I to II CRCs is correlated with a significant reduction in disease-free survival [77]. The product of human colonic bacterium, enterotoxigenic *bacteroides fragilis*, substantially induced CRC onset via STAT3 activation in Th17 cells [78]. In inflammation-associated colon cancer, increased TLR4 expression in intestinal epithelial cells leads to the activation of STAT3, which promotes the growth of CRC in vivo [21]. Furthermore, TLR4/STAT3 signaling has been shown to correlate with the clinical stage in human colorectal adenocarcinoma [21].

### 2.3. Prostate Cancer

The IL-6 levels in serum are increased in patients with castration-resistant or untreated metastatic prostate cancer and are associated with poor outcomes and resistance to treatment with chemotherapy [79]. Serum sIL6R levels have also been shown to be associated with the progression and metastasis of prostate cancer [79]. Serum IL-11 is a potential tumor biomarker for advanced prostate cancer [80]. The augmented expression of IL-11R and the activation of STAT3 have been observed in human prostate cancer [81,82], indicating IL-11R as a promising therapeutic target against human androgen-resistant and advanced prostate cancer [82]. However, the activation status of STAT3 has been reported to be inversely associated with the progression of distant metastases in prostate cancer [83], whereas conflicting reports suggest that it is an effective prognostic marker for prostate cancer [84]. Therefore, further evaluation is warranted. Furthermore, IL-6/STAT3 signaling has been implicated in the conversion from androgen-sensitive to androgen-resistant prostate cancer via the recruitment of myeloid-derived suppressor cells (MDSCs) [85,86]. Using a mouse model of prostate cancer, androgen deprivation has been shown to activate nuclear factor-κB (NF-κB) and STAT3 signaling in prostate cancer cells via leukocyte infiltration, which triggers androgen-dependent tumor cell death and consequently promotes androgen-independent survival. However, cytokines that exclusively activate STAT3 signaling in this environment have not yet been identified [87]. NF-κB activation in prostate cancer cell lines leads to the increased production of IL-6, which contributes to docetaxel resistance [88]. Thus, treatment by the simultaneous inhibition of NF-κB and IL-6/STAT3 signaling is a possible therapeutic strategy to improve the response to chemotherapy and radiation in prostate cancer. STAT3 has been demonstrated to directly bind to androgen receptors and to transcriptionally augment androgen-receptor-targeted genes, even upon a lack of high doses of androgen [89]. In contrast, the silencing of androgen receptor expression enhances CSC-like traits in prostate cancer via IL-6/STAT3 signaling [32]. In addition, blocking the JAK-STAT3 axis suppresses tumor onset and the self-renewal of prostate CSC-like cells [90].

### 2.4. Breast Cancer

Serum IL-6 levels in breast cancer patients have been reported to correlate with a poor prognosis and metastasis [91,92]. In contrast, local intratumoral autocrine/paracrine IL-6 signaling is crucial for regulating breast cancer cell proliferation, metastasis, and cancer stem cell self-renewal [93,94]. Augmentation of the IL-6-meditaed inflammatory loop induces resistance to trastuzumab, a HER2-targeted therapy used for HER2-positive breast cancer, by expanding the CSC population [95]. IL-6/STAT3 signaling is required for the maintenance of breast CSCs and tumor growth [31]. In particular, the IL-6/STAT3 pathway was found to be preferentially active in CD44^+^CD24^−^ breast cancer cells, which have stem-cell-like characteristics, compared with other tumor cell types, and the inhibition of JAK2 decreased their number and blocked the growth of xenografts [31]. In addition, high levels of IL-6 were associated with resistance to paclitaxel in patients with malignant breast cancer [96]. However, the activation of STAT3 does not appear to be an independent marker of breast cancer prognosis [97,98]. Notably, the upregulated expression of IL-11 and the gp130-STAT3 pathway are implicated in the bone metastasis of breast cancer [99]. Although the activation of the IL-6/STAT3 signal has been primarily identified as being necessary for the proliferation of several types of CSCs, tumor-derived erythropoietin, mainly released under hypoxic conditions, also activates the JAK2–STAT3 axis in breast CSCs and promotes self-renewal [100].

### 2.5. Head and Neck Cancer

Increased expression levels of IL-6 and its receptor have been shown to contribute to poor prognosis in patients with head and neck cancer (HNSCC) [101,102]. Consistent with these findings, STAT3 signaling was found to be hyperactivated in HNSCC and to lead to poor outcomes, but STAT3 mutations are rarely detected [35,103]. Mutations in protein tyrosine phosphatase receptors (PTPRs), such as PTPRT and PTPRD, appear to frequently occur in HNSCC, indicating one cause of the STAT3 hyperactivation in HNSCC [104,105]. STAT3 signaling is a crucial pathway for the regulation of gene expression that promotes cell proliferation and survival as well as for the expression of growth factors and cytokines (such as IL-6, IL-10, VEGF, and TGFβ) that drive immune suppression [35].

EGFR, which acts upstream of the STAT3 signaling pathway, is overexpressed in 80% to 90% of HNSCC tumors and is linked to an overall decrease in survival and progression-free survival [106,107]. This finding led to the approval of the anti-EGFR monoclonal antibody cetuximab for the treatment of HNSCC. In addition, other receptor tyrosine kinases, such as HER2 and MET, are overexpressed in HNSCCs, and their overexpression may be associated with the resistance of HNSCCs to EGFR-targeted drugs that act via the activation of STAT3 and its gene targets [108,109,110].

To date, it has been well-documented that EMT is commonly involved in the acquisition of invasiveness and metastatic potential in malignant HNSCC tumors [111,112]. Mechanistically, IL-6 induces EMT changes in HNSCC cells via the activation of STAT3 signaling [113]. Additionally, cytokines and growth factors in the TME, particularly IL-6, EGF, and hepatocyte growth factor, suppress anoikis by activating tumor cell signaling pathways, including the RAS-MAPK, PI3K-mechanistic target of rapamycin kinase, and STAT3 pathways [114,115,116]. Notably, anoikis suppressors in the TME are produced by infiltrating immune cells, CAFs, endothelial cells, and tumor cells themselves [114], suggesting highly complicated crosstalk between the various cell types that contribute to metastasis in HNSCC.

### 2.6. Lung Cancer

Lung cancer is the leading cause of cancer-associated deaths worldwide, and the most common type of lung cancer is non-small-cell lung cancer (NSCLC), accounting for 85% of all lung cancer cases [117]. The STAT3-activating cytokine IL-6 is upregulated in the serum and exhaled breath condensate of NSCLC patients and correlates with a higher risk of metastasis and chemotherapy resistance [118,119,120,121,122,123,124,125]. Increased IL-11 expression has also been detected in the serum, tumors, and exhaled breath condensate of NSCLC patients and is associated with a higher risk of metastasis [126,127]. A high expression level of OSM is associated with poor outcomes in patients with NSCLC and enhances the EMT of NSCLC cells [128]. In addition, a sustained activation of STAT3 occurs in more than 50% of NSCLC patients [129,130], and its increased expression leads to low-grade tumor differentiation, lymph node metastasis, clinical stage progression, and drug resistance [131,132,133]. Mutations in receptor tyrosine kinases, such as EGFR, and Src family proteins have been associated with the constitutive activation of STAT3 in NSCLC [133,134], and STAT3 activation has been associated with lymph node metastasis and clinical stage progression and is an independent prognostic factor of NSCLC [135,136]. To date, the tumor-promoting functions mediated by STAT3 signaling in NSCLC have been well-documented to promote cell survival, angiogenesis, drug resistance, cancer cell stemness, and cancer immune evasion [117]. As a result, highly increased STAT3 expression enhanced the proliferation, survival, and radioresistance of NSCLC cells [132], whereas dominant-negative STAT3 resulted in the suppression of human lung cancer cell proliferation and invasive potential [137].

JAK-STAT3 signaling occurs during the early adaptive response to EGFR-tyrosine kinase inhibitor (TKI) therapy in EGFR-mutant NSCLC and may occur together with the downstream signaling of NF-κB activation [138]. In preclinical NSCLC models, such as patient-derived tumor xenograft models and cell lines, response rates to EGFR TKI therapy were improved by the addition of JAK or STAT3 inhibitors [138,139,140,141]. IL-6 autocrine signaling by tumor cells enhanced the activation of the JAK-STAT3 signaling pathway, whereas the addition of neutralizing anti-IL-6 antibodies reduced tumor growth in a mouse model [134,142]. Nevertheless, early clinical trials showed only a 5% response rate to treatment with the JAK inhibitor ruxolitinib in combination with erlotinib in patients who showed cancer progression during their prior treatment with erlotinib, suggesting that treatment with these drug combinations is not able to reverse previously established drug resistance [143]. Because early adaptive activation of JAK-STAT3 signaling was observed in preclinical models in response to EGFR TKI treatment [134,141], a combination of a JAK and/or STAT3 inhibitor and an EGFR TKI may be necessary for therapeutic efficacy [133,137]. Therefore, the JAK inhibitor INCB39110 has been investigated for its use as a treatment in combination with the third-generation EGFR KI osimertinib in patients with the EGFR-T790M mutation, which is a secondary site mutation in which methionine is substituted for threonine at position 790 that is found in more than 50% of patients with acquired resistance to EGFR TKIs, such as erlotinib and gefitinib. [144]. The coactivation of STAT3 and Yes1 associated transcriptional regulator (YAP1) has also been associated with the promotion of tumor cell survival after EGFR TKI treatment, and the co-inhibition of EGFR, STAT3, and Src-YAP1 signaling demonstrates a more effective synergistic effect than the single use of an EGFR TKI [139].

## 3. STAT3 in Cells of the TME

The TME of most cancers is rich in immune cells, immunosuppressive, and often affected by the complex immunomodulatory actions of IL-6 family cytokines. IL-6/STAT3 signaling in Th1 cells has the most obvious effect on the TME by suppressing cell-mediated antitumor immunity, whereas chronic inflammation contributes to the promotion of tumor progression and the dysregulation of angiogenesis and affects the recruitment, retention, and infiltration of leukocytes as well as immune responses via the activation of IL-6/STAT3 signaling in multifaceted innate and adaptive immune cells and nonimmune cells, such as cancer-associated fibroblasts and endothelial cells. In particular, during tumor initiation, the IL-6/STAT3 signal promotes the generation of pathogenic Th17 cells and MDSCs, suppressing antigen-presenting dendritic cells and antitumor cytotoxic CD8^+^ T cells and promoting regulatory T (Treg)-cell activity and tumor-associated macrophage phenotype switching from tumorigenic M1-type traits to immunosuppressive M2-type traits. Similarly, IL-11/STAT3 signaling facilitates inflammation-associated tumorigenesis in the gastrointestinal tract and polarizes T cells and macrophages to a more immunosuppressive phenotype [11,145,146,147]. In addition to its effects on tumor-associated immune cells, the enhanced activity of STAT3 via IL-6 family cytokines on CAFs is of great interest regarding its indirect tumor-promoting effects (Figure 1).

### 3.1. Immune Cells

STAT3 promotes immunosuppressive effects on the functions of CD8^+^ effector cells. The ablation of STAT3 in hematopoietic cell lineages facilitates antitumor immunity to inhibit tumor proliferation and metastasis via enhancing the functions of CD8^+^ T cells, natural killer (NK) cells, dendritic cells, and neutrophils in a murine model of melanoma, suggesting that STAT3 signaling may suppress tumor immune surveillance systems [148]. In addition, in a mouse transplant model, a loss of STAT3 signaling in the hematopoietic compartment facilitated the recruitment of tumor-infiltrating effector T cells and attenuated the infiltration of Treg cells. Consistently, the activation of STAT3 has been shown to be crucial for restricting the recruitment and activation of CD8^+^ T cells that are required to prevent the progression of melanoma [149,150]. Similarly, constitutively active STAT3 in human CD4^+^ T cells suppresses antitumor immunity by blocking the production of granzyme B, tumor necrosis factor (TNF), interferon gamma (IFNγ), IL-13, and other inflammatory cytokines [151]. 

IL-6/STAT3 signaling in CD4^+^ T cells is crucial for the differentiation of Th17 cells via the expression of RAR-related orphan receptor gamma [152,153,154]. Th17 cells comprise approximately 5% of the CD4^+^ T cells in PDACs. The role of Th17 cells in the TME is also context-dependent. In PDAC, IL-17 secretion from γδ T cells and Th17 cells may enhance antitumor immunity [155]. However, early in PDAC carcinogenesis, IL-17 has a direct mitogenic effect on *KRAS*-mutation-induced PanIN cells expressing IL-17R [156]. Although the effects of distinct T-cell subsets depend on the underlying immune context of the tumor due to various physiological conditions and environments and may be altered during the tumor progression of PDAC, the regulation of the differentiation and function of T cells in PDAC TMEs plays a crucial role in tumor immunity.

STAT3 has been demonstrated to be a transcriptional activator of the immune checkpoint molecules cytotoxic T-lymphocyte-associated protein 4 (CTLA4), programmed cell death 1 (PDCD1, also known as PD-1), and CD274 (also known as PD-L1) in T cells. Consistently, the promoter region of the *PD-1* gene contains STAT3 binding sites, and PD-1 expression is promoted in response to signaling via the T-cell receptor/nuclear factor of activated T cells, IL-6/STAT3, and IL-12/STAT4 [157]. Notably, it has been demonstrated that PD-1 signaling is activated via STAT3 on CD4^+^ T cells and promotes collagen production by fibroblasts in pulmonary fibrosis, indicating the role of STAT3 in immunosuppression and tumor-promoting responses in the TME [158].

It has been well-documented that IL-6 inhibits the TGF-β-induced generation of Treg cells [159] via the IL-6/STAT3-mediated direct suppression of *forkhead box P3* (*FOXP3*), which is a key transcriptional regulator of Treg cell differentiation on naive T cells [160]. Importantly, these inhibitory effects of IL-6/STAT3 signaling are restricted to inducible Treg cells and have no effects on the differentiation and function of natural Treg cells [161]. Interestingly, in the TME, IL-10-mediated STAT3 activation promotes Treg cell differentiation and enhances CTLA4 expression [162]. STAT3 can also activate *FOXP3* gene expression, resulting in the promotion of Treg differentiation [163,164]. Therefore, STAT3 can regulate the peripheral immunity and tolerance of effector T cells in the TME, indicating that STAT3 is a potential therapeutic target to suppress the formation of an immunosuppressive TME.

The existence of M2 macrophages in the TME has been shown to be associated with poor outcomes in patients with most types of solid tumors [165]. Excess STAT3 activation promotes the polarization of M2 macrophages and increases the expression levels of *arginase-1* (*ARG-1*), *Fos-related antigen 1*, *TGF-β*, *IL-10*, and *VEGF-a*, which are M2-associated markers [166]. M2-polarized macrophages have been implicated in the promotion of tumor growth of melanoma and Lewis lung cancer [166]. M2 macrophages can also facilitate STAT3 signaling in breast cancer cells to promote tumor proliferation [167], and STAT3 activation in M2-polarized macrophages can activate STAT3 signaling in ovarian cancer cells to promote their growth via the production of IL-6 and IL-10 [168]. In the TME, the Th2 cell cytokine IL-4 can also be activated by macrophages to promote their growth and induce STAT3-dependent cathepsin secretion by macrophages, supporting the development of pancreatic neuroendocrine tumors [169]. The TLR-induced increase in the activation of STAT3 has been shown to increase PD-L1 levels in M2 macrophages [170].

MDSCs, which are derived from pathologically activated neutrophils and monocytes, have been implicated in the promotion of immunosuppressive effects on antitumor immune cells in the TME [171]. STAT3 promotes the differentiation and expansion of MDSCs, thereby enhancing their ability to suppress effector T cells and promote the differentiation of Treg cells, which enable the augmentation of tumor formation [172]. The knockdown of STAT3 in MDSCs derived from patients with prostate cancer reduced the immunosuppressive functions of MDSCs against effector T cells by STAT3 [173]. Notably, a mouse model of acute colitis bearing the Y757F point mutation in murine gp130 (gp130^Y757F/Y757F^), which abrogates the SOCS3- and/or SHIP2-mediated negative feedback loop of the IL-6/STAT3 signal, resulting in the hyperactivation of STAT3 and has been shown to be resistant to colorectal damage and weight loss. This effect was shown to be produced by a small number of STAT3-induced granulocytic MDSCs (gMDSCs; also known as polymorphonuclear MDSCs [171]) with high expression of *Arg1* and anti-inflammatory Th2 cytokines, such as *IL-10* and *TGF-β*, [174], indicating that STAT3 promotes a precancerous host defense response during ulcerative colitis. In addition, gMDSCs are enriched by the local presence of IL-11, which activates STAT3 in CRC [175].

To limit the destructive responses of neutrophils, STAT3 necessarily acts as a negative regulator of neutrophil functions, suppresses the production of antitumor Th1 cytokines, such as IL-1, TNF, and IFNγ [176], and causes the unresponsiveness of neutrophils to chemotaxis by CXC-chemokine receptor 2 ligands [177,178]. Although neutrophils are necessary to limit the destructive inflammatory effects on the host, they may also facilitate tumorigenesis. In fact, the abrogation of STAT3 in neutrophils enhanced the cytolytic activity of neutrophils and induced tumor regression [148]. Similarly, STAT3 deletion in NK cells increased the expression of cytotoxic factors, such as perforin and granzyme B, and the activation of the NK cell marker CD226 [179]. On the other hand, the mutation-driven excessive STAT3 activity in NK cells from patients with chronic lymphoproliferative disorders of NK cells, T-cell large granular lymphocytic leukemias [180], and NK/T-cell lymphomas of the nasal type [181] promotes lymphomagenesis and provides a tumor-promoting TME. Thus, it is clear that the dysregulation of STAT3 in innate immune cells augments cancer cell proliferation and inhibits antitumor responses via the immunosurveillance system.

### 3.2. Non-Immune Cells 

Fibroblasts exist in every solid organ to maintain their morphology and function by depositing extracellular matrix proteins and secreting soluble factors [182]. Histological similarities, such as mesenchymal morphology, are maintained among fibroblasts in various organs, but their genomic landscapes differ depending on the organs in which they are located [183]. It has been demonstrated that some fibroblasts contribute to tumor initiation, progression, and metastasis [184]. STAT3 signaling in CAFs [185] and tumor cells [185,186] may induce stromal remodeling of the TME characterized by fibrogenesis, a dysregulated organization of the ECM, and fibroblast contractility, which promote tumor cell motility, invasive activity, and resistance to chemical and immunological therapies [187]. However, the constitutive activation of STAT3 via the Y757F point mutation in pg130 in a mouse pulmonary fibrosis model not only promotes fibrosis in the absence of the TGF-β signaling molecule SMAD3, which is well-known to be crucial in the pathogenesis of lung fibrosis [188], but also results in desmoplasia formation and epithelial stiffness, which enhance tumorigenesis in PDAC mouse models [186]. CAFs assist tumor growth and dissemination through the production of factors such as EGF, IL-6, TGF-β, and VEGF, which promote tumor cell proliferation and angiogenesis [189,190]. The protumorigenic characteristics of these fibroblasts are partially modulated by STAT3 activity via the induction of various cytokines, including LIF [191]. For example, STAT3 acts as a downstream signaling molecule of the focal adhesion kinase-Src-JAK2 axis in CAFs, leading to the expression of CCL2 and immunosuppression. Mechanistic studies have shown that CAFs promote the growth of murine hepatocellular carcinomas by resulting in the mobilization of MDSCs [192]. Factors produced by CAFs can promote STAT3 signaling in other cell types, support intercellular communication between immune cells within the TME, and induce immunosuppression [193,194].

## 4. Promising Target Molecules in STAT3-Associated Tumors

### 4.1. STAP-2

Signal-transducing adaptor protein-2 (STAP-2) was originally identified as a c-Fms/macrophage colony-stimulating factor receptor-binding protein containing pleckstrin-homology (PH), SH2, and proline-rich domains. Interestingly, STAP-2 levels were strongly induced in the liver by lipopolysaccharide (LPS) stimulation and in isolated hepatocytes by IL-6 stimulation. Consistently, in STAP-2-deleted hepatocytes, the acute phase responses induced by IL-6 and the tyrosine phosphorylation levels of STAT3 were significantly decreased. Moreover, STAP-2 binds to activated STAT3 via a YXXQ motif in the C-terminal region, indicating that STAP-2 is an adaptor molecule that modulates STAT3 activity [195].

Breast tumor kinase (BRK, also known as protein tyrosine kinase 6), which is related to the Src family of tyrosine kinases, is overexpressed in approximately 85% of invasive ductal carcinomas [196]. STAP-2, which was the first BRK substrate to be identified [33], binds to BRK via its PH domain and contributes to the activation of STAT3 [197], and as the PH domain of STAP-2 is essential for the translocation of STAP-2 to the plasma membrane after EGF treatment, the binding ability of the PH domain of STAP-2 to BRK is considered to be an important biological characteristic [33]. Thus, the PH domain of STAP-2 may have a biological function in altering the subcellular localization of BRK and promoting its activation; STAP-2 acts as a scaffold protein that facilitates the interaction between BRK and STAT3. Taken together, the experiments using deletion mutants suggest that STAP-2 modulates multiple events, i.e., the binding of STAP-2 to BRK, the activation of BRK, and the subsequent promotion of the tyrosine phosphorylation of STAT3. Thus, STAP-2 functions in concert with BRK to promote breast cancer cell proliferation. As both BRK and STAP-2 expression are high in breast cancer cells, their coupling may promote the abnormal activation of STAT3. These data may provide insights into the molecular mechanisms and implications of the BRK/STAP-2/STAT3 interaction and may provide clues for the development of novel therapies for breast cancer.

STAP-2 enhances signaling via EGFR through its protein stabilization, leading to higher tumorigenesis in prostate cancer cells [198]. STAP-2 promotes EGFR signaling through a two-step process: the stabilization of EGFR and the subsequent activation of STAT3 through their direct interaction. In addition to EGFR, IL-6 signaling also activates STAT3, and IL6R blockade significantly inhibits tumor progression [199]. A knockdown of STAP-2 may inhibit prostate cancer cell growth by synergistically inhibiting EGFR and IL6R signaling. As described above, STAP-2 can bind to BRK and contributes to promoting the BRK-modulated activation of STAT3 and STAT5 [197,200]. In particular, BRK augments EGFR signaling by decreasing the casitas B-lineage lymphoma (CBL)-enhanced ubiquitination of EGFR [201]. The expression of cell-surface EGFR is important for the activation of RAS and ERK, and STAP-2 prevents the reduction in surface expression levels of EGFR, even after EGF treatment. Another biological mechanism of STAP-2 in prostate cancer cells is to inhibit the CBL-enhanced ubiquitination of EGFR and to restore EGFR [198]. A knockdown of STAP-2 inhibits the cell growth of prostate cancer cells [198]. STAP-2 is also a potent regulator of EGFR activation in prostate cancer cells. STAP-2 does not bind to EGFR K721A, a dimer-formation-deficient mutant, suggesting that STAP-2 interacts with EGFR after dimer formation [198]. Furthermore, STAP-2 acts to stabilize wild-type EGFR after EGF stimulation but not inactive EGFR mutants [198]. Indeed, the inhibition of tumor cell growth by STAP-2 knockdown occurs under EGFR-activated conditions but not under EGFR-inactivated conditions [198]. Notably, gefitinib treatment fails to further inhibit the cell proliferation of STAP-2-silenced prostate cancer cells [198]. The distinct regulatory mechanisms for EGFR surface expression in gefitinib-treated and STAP-2 knockdown cells demonstrates that the inhibition of STAP-2 function can destabilize both wild-type EGFR and gefitinib-resistant self-activated EGFR [198]. Therefore, STAP-2 inhibitors may have the potential to be effective anticancer agents against gefitinib-resistant prostate cancer (Figure 2).

### 4.2. ARID5A

Recent studies have demonstrated the crucial roles of Arid5a in inflammation, autoimmunity, and cancer [34,202]. Arid5a was identified as an RNA-binding protein that directly binds to a stem–loop element in the 3′-untranslated regions (UTRs) of *IL-6* to stabilize *IL-6* mRNA and augments IL-6 expression. Stimulation with LPS, IL-1β, or IL-6 leads to its expression in macrophages and embryonic fibroblasts. Arid5a has been shown to exacerbate symptoms in LPS-treated mice and experimental autoimmune encephalomyelitis (EAE) mice, accompanied with an increase in IL-6 levels [203]. The binding element of Arid5a in the *IL-6* 3′ UTR coincides with that of Regnase-1, resulting in Arid5a counteracting the destabilizing function of Regnase-1 on *Il6* mRNA. Notably, in untreated rheumatoid arthritis (RA) patients, the expression of ARID5a in CD4^+^ T cells is enhanced, whereas treatment with the anti-IL6R antibody tocilizumab results in a decrease in the expression of Arid5a [204], indicating that the IL-6-ARID5A axis may be involved in the pathogenesis of RA. Importantly, Arid5a is deeply involved in the Th17-polarized differentiation of naïve CD4+ T cells via the stabilization of *STAT3* mRNA, resulting in the development of EAE in mice [205].

Recently, the expression levels of Arid5a have been reported to be significantly increased in mesenchymal tumor subtypes of PDAC and CRC, such as the quasi-mesenchymal and consensus molecular subtype 4 subtypes, respectively [206]. In cells derived from the PDAC mouse model KPC (Pdx1-cre, KRASG12D, and p53R172H), the abrogation of Arid5a was shown to significantly downregulate EMT-TFs and EMT markers, such as *Zeb1*, *Zeb2*, *Snai1*, *Snai2*, *Twist2*, *Acta2*, and *Itgb1*, compared with WT KPC cells, whereas the expression of the representative epithelial marker E-cadherin (*Cdh1* gene) was substantially increased [206]. Additionally, an ingenuity pathway analysis demonstrated that signaling pathways linked to EMT and metastasis, including the regulation of EMT by growth factors/development, IL-8, OSM, and stemness signals, were diminished upon the loss of *Arid5a*. In addition, the signaling pathways of IL-6, STAT3, and JAK/STAT were downregulated in Arid5a-deficient KPC cells [206]. Moreover, Arid5a expression was enhanced in in vitro EMT models induced by IL-6 and TGF-β stimulation [206], indicating the involvement of Arid5a in inducing the mesenchymal cell properties of PDAC. In agreement with these findings, a recent report indicated that the IL-6-Arid5a axis enhances the invasive and metastatic activities of breast cancer cells. Mechanistically, the Arid5a induced by IL-6 functions as a transcription factor, increasing the expression levels of the long non-coding RNA AU021063. Subsequently, AU021063 functions to activate breast cancer cell invasion and metastasis through the stabilization of *tribbles homolog 3* mRNA and the activation of the Mek/Erk signaling pathway [207].

It has been shown that Arid5a enables mesenchymal tumor models of PDAC and CRC to facilitate immune evasiveness via promoting the tumor infiltration of immunosuppressive granulocytic MDSCs and Tregs and reducing the recruitment and activation of antitumor effector T cells [206]. Mechanistically, Arid5a functions as a dual regulator, leading to the formation of immunosuppressive TMEs in malignant tumors and triggering the metabolic reprograming and recruitment of suppressive immune cells. First, Arid5a promotes the inhibitory effect of Ido1 on effector CD4^+^/CD8^+^ T cells via the stabilization of *Ido1* mRNA by binding to its 3′-UTR and by reducing the intratumoral tryptophan concentration [208,209]. Additionally, Ido1 expression in tumor tissues promotes Treg differentiation/activation by catabolizing tryptophan to produce kynurenine and ultimately activating aryl hydrocarbon receptors (AhR) [210,211]. AhR activation results in the extensive infiltration of gMDSCs to the TME [212]. Second, Arid5a upregulates chemokine Ccl2 expression in the TME via the stabilization of its mRNA and then Ccl2 enhances the infiltration of immunosuppressor cells, such as Tregs and gMDSCs [213,214,215,216], to the TME [206] (Figure 3).

## 5. Perspectives

In tumorigenesis, chronic inflammation and metabolic changes associated with genetic mutations in normal cells enable transformed cells to escape the immunological defense mechanisms of the tissue and to reprogram the functions of endogenous signaling mechanisms and cell populations in surrounding tissues, thereby disrupting the homeostatic balance of the entire organism and creating neoplasms within the organism. STAT3 plays a central role in this entire process. To date, a large amount of effort has been put into developing STAT3 inhibitors that both directly and indirectly target STAT3, including SH2 domain inhibitors and DNA-binding domain inhibitors, and JAK kinase inhibitors and Src inhibitors, respectively, and integrating STAT3-based combination immunotherapies [12,35] (Figure 1, Table 1 and Table 2). In particular, since its approval in 2009, tocilizumab has been used to inhibit IL-6/STAT3 signaling in autoimmune diseases, such as rheumatoid arthritis, caused by the overexpression of IL-6 and acute inflammatory diseases caused by cytokine storms associated with chimeric antigen receptor T-cell therapy and SARS-CoV-2 infection, and it has shown high therapeutic efficacy against various immune diseases. On the other hand, few effective therapies that target STAT3 signaling for the treatment of cancer in clinical practice have been developed [6,7,8,9,10] (Table 1 and Table 2). As mentioned above, cancer is a complex interplay of diverse cell populations that result in malignant transformation. Therefore, whereas analyses of the expression and function of molecules associated with STAT3 activation can assess the local and steady-state malignant potential of a cancer, they are insufficient to predict the stage or detailed course of a cancer. Furthermore, it is clear that in addition to STAT3 signaling various groups of molecules are involved in cancer development. The mode of interaction between these molecules also needs further study. In the future, it will be essential to perform spatiotemporal gene expression analyses to analyze multiple cell populations, improve technologies for the detection of aging and inflammation using artificial intelligence, and introduce mathematical analysis technology to integrate these technologies. Furthermore, it is also necessary to enhance the convergence of life science, physical science, engineering science, and computational science to create the next generation of cancer therapeutics.

## Figures and Tables

**Figure 1 cells-11-02618-f001:**
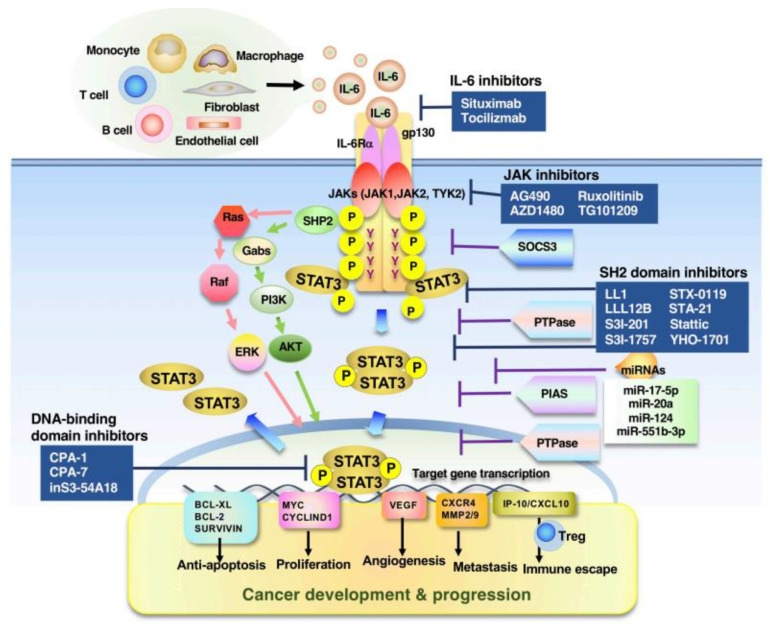
Multifaceted roles of STAT3 in physiological and pathological processes, and inhibitors targeting STAT3 signaling. STAT3 is activated by specific cytokines, growth factors, etc., and contributes to multiple physiological functions by regulating its target genes as a transcription factor. Representative STAT3 inhibitors are classified as those that target STAT3 directly (e.g., SH2 domain and DNA-binding domain inhibitors) and indirectly (e.g., JAK kinase and IL-6 inhibitors).

**Figure 2 cells-11-02618-f002:**
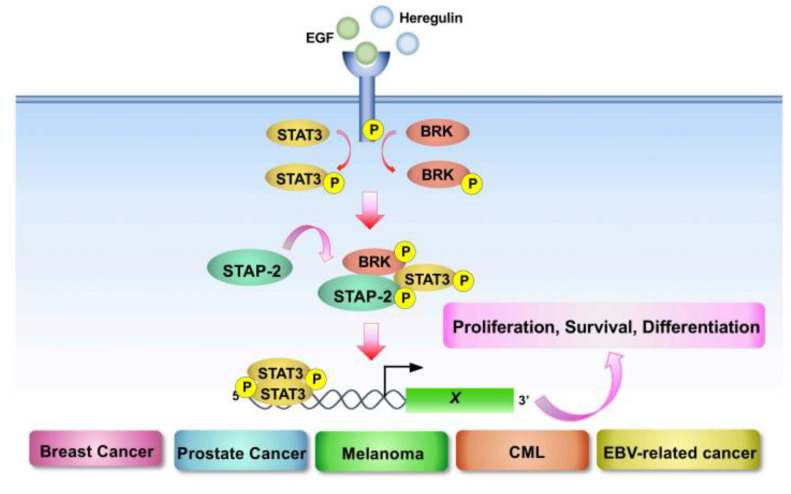
Functional roles of STAP-2 as a promising target molecule in STAT3-associated tumors. Stimulation by EGF or other molecules induces the phosphorylation of STAT3 and BRK, and STAP-2 is also phosphorylated and interacts with STAT3 and BRK as a scaffold protein. Subsequently, STAT3 translocates to the nucleus, where it regulates target genes and contributes to proliferation, etc.

**Figure 3 cells-11-02618-f003:**
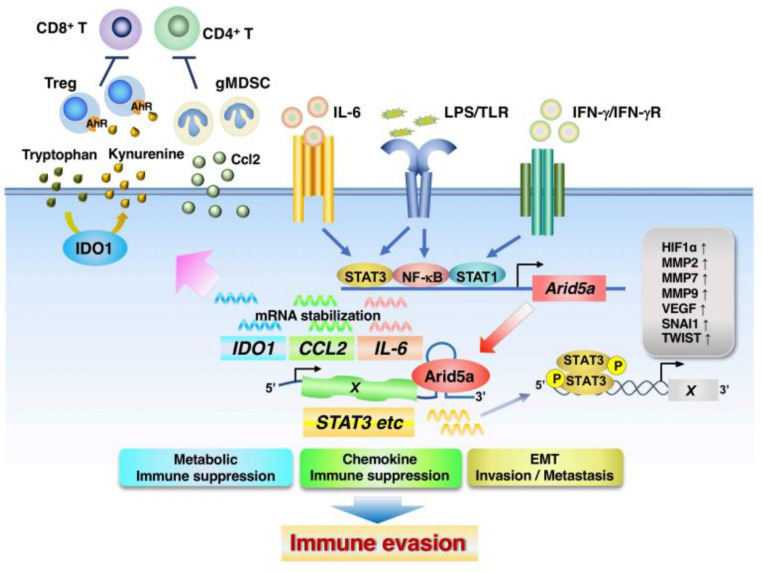
Arid5a mediates immune evasion. Arid5a is induced by LPS, IL-1β, or IL-6, directly binds to a stem–loop element in the 3′-untranslated regions of target genes to stabilize their mRNA, and augments their expression. Arid5a induces immune evasion by contributing to metabolic reprogramming, the upregulation of immunosuppressive chemokines, and the induction of mesenchymal properties through RNA stabilization.

**Table 1 cells-11-02618-t001:** Preclinical studies on STAT3 inhibitors.

Action	Inhibitor/ Compound	Mechanism of Action	Cancer Type	Ref.
Direct inhibitors	LL1	SH2 domain inhibitor	CRC, NSCLC	[217]
	LLL12B	SH2 domain inhibitor	Medulloblastoma	[218]
	S3I-201	SH2 domain inhibitor	Breast cancer, liver cancer	[219]
	S3I-M2001	SH2 domain inhibitor	Breast cancer	[220]
	S31-1757	SH2 domain inhibitor	Breast cancer, lung cancer	[221]
	STX-0119	SH2 domain inhibitor	Glioblastoma	[222]
	STA-21	SH2 domain inhibitor	Breast cancer	[223]
	Stattic	SH2 domain inhibitor	Breast cancer, HNSCC	[224]
	YHO-1701	SH2 domain inhibitor	HNSCC, NSCLC	[225]
	PY*LKTK	SH2 domain inhibitor	NIH3T3/v-Src or v-Ras	[226]
	CPA-1	DNA-binding domain inhibitor	Breast cancer, colon cancer, melanoma	[227]
	CPA-7	DNA-binding domain inhibitor	Prostate cancer, breast cancer, colon cancer, melanoma	[227,228]
	inS3-54A18	DNA-binding domain inhibitor	NSCLC	[229]
	DBD-1	DNA-binding domain inhibitor	Melanoma, myeloma	[230]
Indirect inhibitors	AG490	JAK inhibitor	Ovarian cancer, pancreatic cancer	[231]
	AZD1480	JAK inhibitor	Lymphoma, lung cancer	[232,233]
	Ruxolitinib	JAK inhibitor	Hepatocellular carcinom	[234]
	TG101209	JAK2 inhibitor	Leukemia	[235]
	WP1066	JAK inhibitor	Renal cell carcinoma	[236]
	KDI1	RTK inhibitor	Vulval and breast cancer	[237]
	PD153035	RTK inhibitor	Oral squamous carcinoma	[238]
	Dasatinib	Src inhibitor	Synovial sarcoma, hepatocellular carcinoma, glioma, prostate cancer	[239]

Abbreviations: CRC—colorectal cancer; HNSCC—head and neck squamous cell carcinoma; NSCLC—non-small cell lung carcinoma.

**Table 2 cells-11-02618-t002:** STAT3 inhibitors being tested in clinical trials.

Action	Inhibitor/Compound	Type	Cancer Type	Phase	NCT Number
Direct inhibitors	BBI608 (FDA approved)	Small molecules	Advanced malignancies	I/II	NCT01775423
			CRC	III	NCT01830621
	C188-9	Small molecules	BC, CRC, HNSCC, HCC, NSCLC, GAC, melanoma, advanced cancer	I	NCT03195699
	OPB-31121	Small molecules	advanced cancer, solid tumorS	I	NCT00955812
			HCC	I/II	NCT01406574
	OPB-51602	Small molecules	Malignant solid tumors	I	NCT01184807
			Hematological malignancies	I	NCT01344876
			Nasopharyngeal carcinoma	I	NCT02058017
	OPB-111077	Small molecules	Acute myeloid leukemia	I	NCT03197714
			Advanced HCC	I	NCT01942083
	AZD-9150	Oligonucleotides	Lymphoma	I/II	NCT01563302
Indirect inhibitors	AZD-1480	JAK1/2	Solid tumors	I	NCT01112397
	CYT387	JAK1/2	Myelofibrosis	I/II	NCT02101268
			PMF, post-PV, post-ET MF	III	NCT03427866
	Ruxolitinib (FDA approved)	JAK1/2	Myelofibrosis	III	NCT03427866
	LY2784544	JAK2	Myeloproliferative neoplasms	II	NCT01594723
	SB1518	JAK2	Myelofibrosis	III	NCT02055781
	Siltuximab (FDA approved)	IL-6R	Multiple myeloma	II	NCT03315026
	Tocilitizumab (FDA approved)	IL-6R	HCC	I/II	NCT02997956
Combinations	AZD9150, durvalumab (anti-PD-L1)	Direct inhibitors and ICB	NSCLC	II	NCT03334617
			PC, CRC, NSCLC	II	NCT02983578
			Advanced solid tumors, metastatic HNSCC	I/II	NCT02499328
			Diffuse large B-cell lymphoma	I	NCT02549651
	BBI608, nivolumab (anti-PD-1)	Direct inhibitors and ICB	Metastatic CRC	II	NCT03647839
	BBI608, pembrolizumab (anti-PD-1)	Direct inhibitors and ICB	Metastatic CRC	I/II	NCT02851004
	Apatinib, SHR-1210 (anti-PD-1)	Indirect inhibitors and ICB	Melanoma	II	NCT03955354
	Bevacizumab, atezolizumab (anti-PD-L1)	Indirect inhibitors and ICB	Unresectable HCC	III	NCT03434379
	Dasatinib, Ipilimumab (anti-CTLA-4)	Indirect inhibitors and ICB	GIST, stage III/IV soft tissue sarcoma	I	NCT01643278
	Dasatinib, nivolumab (anti-PD-1)	Indirect inhibitors and ICB	Philadelphia chromosome positive ALL	I	NCT02819804
	Ruxolitinib, pembrolizumab (anti-PD-1)	Indirect inhibitors and ICB	Hematological malignancies	II	NCT04016116
			Metastatic stage IV TNBC	I	NCT03012230

Abbreviations: ALL—acute lymphoblastic leukemie; BC—breast cancer; CRC—colorectal cancer; GAC—gastric adenocarcinoma; GIST—gastrointestinal stromal tumor; HCC—hepatocellular carcinoma; HNSCC—head and neck squamous cell carcinoma; NSCLC—non-small cell lung carcinoma; PC—pancreatic cancer; PMF—primary myelofibrosis; Post-PV—post polycythemia vera; Post-ET MF—post-essential thrombocythemia myelofibrosis; TNBC—triple negative breast cancer.

## Data Availability

Not applicable.

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
