# Peer review of "Central Roles of STAT3-Mediated Signals in Onset and Development of Cancers: Tumorigenesis and Immunosurveillance"

_cells, 2022, doi:10.3390/cells11162618_

Round 1
Reviewer 1 Report
In the paper by Alsaeed et al., Hascimoto S. et al described the role of STAT3 in cancerogenesis and tumor processes. The authors summarised the impact of this gene in different kind of cancers, in the tumor microenvironment and its promising role as therapeutic target.
The manuscript looks like well written and organized. The authors have presented an interesting topic in the field of oncology. The paper should be considered after major revisions.
The manuscript would benefit from the following:
1. The study reports the role of IL6/STAT3-mediated signaling in prostate, pancreatic, colorectal and breast cancers. These paragraphs increase the impact of the review in the field of cancer research but the kind of tumors included should be increased. In particular, the authors should report the implication of STAT3 in cancer development and as possible therapeutic target in the precision medicine approach for lung cancers (DOI: 10.3390/cancers12051107, DOI: 10.3390/cancers13246228, DOI: 10.1016/j.semcancer.2020.07.009) and head and neck cancers (these references should be included: DOI: 10.1016/j.oraloncology.2015.11.022, DOI: 10.1172/JCI137001, DOI: 10.1016/j.prp.2020.153172, DOI: 10.3390/jpm12060854).
2. Figure 3 is not really clear. For example, the immune cells and the arrows disposition result chaotic and some elements have a low resolution. The authors should organize differently the elements and improve the figure resolution.
Author Response
We would like to thank the reviewers for their thoughtful comments and helpful suggestions to improve our review manuscript. Outlined below are our point-by-point responses to the comments.
Response to 1:
In accordance with the reviewer’s suggestion, we added paragraphs to summarize the recent advances in our understanding of the roles of STAT3 in lung cancer, and head and neck cancer. We included the following explanations after section 2.4. breast cancer (page 14, line 10 to page 17, line 17) of the revised manuscript, as follows:
“2.5. Head and neck cancer
Increased expression levels of IL-6 and its receptor have been shown to contribute to the poor prognosis in patients with head and neck cancer (HNSCC) [101,102]. Consistent with these findings, STAT3 signaling was found to be hyperactivated in HNSCC, and leads to poor outcomes, but STAT3 mutations are rarely detected [35,103]. Mutations in protein tyrosine phosphatase receptors (PTPRs), such as PTPRT and PTPRD, appear to frequently occur in HNSCC, indicating one cause of the STAT3 hyperactivation in HNSCC [104,105]. STAT3 signaling is a crucial pathway for the regulation of gene expression that promotes cell proliferation and survival, as well as for the expression of growth factors and cytokines (such as IL-6, IL-10, VEGF, and TGFβ) that drive immune suppression [35].
EGFR, which acts upstream of the STAT3 signaling pathway, is overexpressed in 80% to 90% of HNSCC tumors, and is linked to an overall decrease in survival and progression-free survival [106,107]. This finding led to the approval of the anti-EGFR monoclonal antibody cetuximab for the treatment of HNSCC. In addition, other receptor tyrosine kinases, such as HER2 and MET, are overexpressed in HNSCCs, and their overexpression may be associated with the resistance of HNSCCs to EGFR-targeted drugs that act via the activation of STAT3 and its gene targets [108-110].
To date, it has been well documented that EMT is commonly involved in the acquisition of invasiveness and metastatic potential in malignant HNSCC tumors [111,112]. Mechanistically, IL-6 induces EMT changes in HNSCC cells via the activation of STAT3 signaling [113]. Additionally, cytokines and growth factors in the TME, particularly IL-6, EGF, and hepatocyte growth factor, suppress anoikis by activating tumor cell signaling pathways, including the RAS-MAPK, PI3K-mechanistic target of rapamycin kinase and STAT3 pathways [114-116]. Notably, anoikis suppressors in the TME are produced by infiltrating immune cells, CAFs, endothelial cells, and tumor cells themselves [114], suggesting a highly complicated crosstalk between the various cell types that contribute to metastasis in HNSCC.
2.6. Lung cancer
Lung cancer is the leading cause of cancer-associated deaths worldwide, and the most common type of lung cancer is non-small cell lung cancer (NSCLC), accounting for 85% of all lung cancer cases [117]. The STAT3-activating cytokine IL-6 is upregulated in the serum and exhaled breath condensate of NSCLC patients, and correlates with a higher risk of metastasis and chemotherapy resistance [118-125]. Increased IL-11 expression has also been detected in the serum, tumors, and exhaled breath condensate of NSCLC patients, and is associated with a higher risk of metastasis [126,127]. A high expression level of OSM is associated with poor outcomes in patients with NSCLC, and enhances the EMT of NSCLC cells [128]. In addition, sustained activation of STAT3 occurs in more than 50% of NSCLC patients [129,130], and its increased expression leads to low-grade tumor differentiation, lymph node metastasis, clinical stage progression, and drug resistance [131-133]. Mutations in receptor tyrosine kinases, such as EGFR, and Src family proteins have been associated with the constitutive activation of STAT3 in NSCLC [133,134], and STAT3 activation has been associated with lymph node metastasis and clinical stage progression, and is an independent prognostic factor of NSCLC [135,136]. To date, the tumor-promoting functions mediated by STAT3 signaling in NSCLC have been well documented to promote cell survival, angiogenesis, drug resistance, cancer cell stemness, and cancer immune evasion [117]. As a result, highly increased STAT3 expression enhanced the proliferation, survival, and radioresistance of NSCLC cells [132], whereas dominant-negative STAT3 resulted in the suppression of human lung cancer cell proliferation and invasive potential [137].
JAK-STAT3 signaling occurs during the early adaptive response to EGFR-tyrosine kinase inhibitor (TKI) therapy in EGFR-mutant NSCLC, and may occur together with the downstream signaling of NF-κB activation [138]. In preclinical NSCLC models, such as patient-derived tumor xenograft models and cell lines, response rates to EGFR TKI therapy were improved by the addition of JAK or STAT3 inhibitors [138-141]. IL-6 autocrine signaling by tumor cells enhanced activation of the JAK-STAT3 signaling pathway, whereas the addition of neutralizing anti-IL-6 antibodies reduced tumor growth in a mouse model [134,142]. Nevertheless, early clinical trials showed only a 5% response rate to treatment with the JAK inhibitor ruxolitinib in combination with erlotinib in patients who showed cancer progression during their prior treatment with erlotinib, suggesting that treatment with these drug combinations is not able to reverse previously established drug resistance [143]. Because early adaptive activation of JAK-STAT3 signaling was observed in preclinical models in response to EGFR TKI treatment [134,141], a combination of a JAK and/or STAT3 inhibitor and an EGFR TKI may be necessary for therapeutic efficacy [133,137]. Therefore, the JAK inhibitor INCB39110 has been investigated for its use as a treatment in combination with the third generation EGFR KI osimertinib in patients with the EGFR-T790M mutation, which is a secondary site mutation in which methionine is substituted for threonine at position 790, and is found in more than 50% of patients with acquired resistance to EGFR TKIs, such as erlotinib and gefitinib. [144]. Coactivation of STAT3 and Yes1 associated transcriptional regulator (YAP1) has also been associated with the promotion of tumor cell survival after EGFR TKI treatment, and the coinhibition of EGFR, STAT3, and Src-YAP1 signaling demonstrates a more effective synergistic effect than the single use of an EGFR TKI [139].”
Response to 2:
In accordance with the reviewer’s comment, we improved Figure 3 for clarity.
Reviewer 2 Report
In the manuscript the authors have discussed a very well known and important topic, though ARID5a is new in the context. The manuscript is very well written and the authors should consider the following topics.
1. The authors must briefly highlight ARID5a in the abstract.
2. The authors must represent a figure, depicting the upstream transcriptional regulation of STAT3. Fig. 1 is seems very superficial. The readers will look for a more comprehensive and updated figure. Since the topic is not new, the readers will look for new information.
3. In the perspective the authors should highlight the present and future of STAT3 targeting drugs. It will be appreciated, if the authors can add a table listing the important clinical trails and their present status.
Author Response
We would like to thank the reviewers for their thoughtful comments and helpful suggestions to improve our review manuscript. Outlined below are our point-by-point responses to the comments.
Response to 1:
In accordance with the reviewer’s suggestion, we briefly highlighted the roles of ARID5A in tumor progression in the Abstract (page 2, line 16 to page 2, line 21), and Introduction (page 6, line 1 to page 6, line 7) of the revised manuscript, as follows:
Abstract
“We have reported a series of studies aiming towards understanding the molecular mechanisms underlying the proliferation of various types of tumors involving signal-transducing adaptor protein-2 as an adaptor molecule that modulates STAT3 activity, and recently found that AT-rich interactive domain-containing protein 5a functions as a mRNA stabilizer that orchestrates an immunosuppressive TME in malignant mesenchymal tumors.”
Introduction
“We previously performed a series of studies analyzing the roles of signal-transducing adaptor protein-2 (STAP2) in the proliferation of several types of cancer cells via acting as an adaptor molecule that modulates STAT3 activity [33], and recently found that AT-rich interactive domain-containing protein 5A (ARID5A) functions as an RNA-binding molecule that stabilizes mRNAs, such as those of indoleamine 2,3-dioxygenase 1 (IDO1), C-C motif chemokine ligand 2 (CCL2), and STAT3, resulting in the induction of an immunosuppressive TME in malignant tumors [34].”
Response to 2:
In accordance with the reviewer’s comment, we improved Figure 1, by adding information regarding STAT3 inhibitors.
Response to 3:
In accordance with the reviewer’s comment, we briefly highlighted the current status of STAT3-targeted drugs in clinical trials and in the preclinical stage in the Perspective (page 29, line 10 to page 29, line 13) of the revised manuscript, as follows, and added Table 1, listing preclinical trials of STAT3-targeting drugs, and Table 2, listing ongoing clinical trials.
Perspective
“To date, a large amount of effort has been put into developing STAT3 inhibitors that both directly and indirectly target STAT3, including SH2 domain inhibitors and DNA-binding domain inhibitors, and JAK kinase inhibitors and Src inhibitors, respectively, and integrating STAT3-based combination immunotherapies [12,35] (Figure 1, Table1 and 2). In particular, since its approval in 2009, tocilizumab has been used to inhibit IL-6/STAT3 signaling in autoimmune diseases, such as rheumatoid arthritis caused by the overexpression of IL-6, and acute inflammatory diseases caused by cytokine storms associated with chimeric antigen receptor T cell therapy and SARS-CoV-2 infection, and has shown high therapeutic efficacy against various immune diseases. On the other hand, few effective therapies that target STAT3 signaling for the treatment of cancer in clinical practice have been developed [6-10] (Table 1 and 2).”